# Multi-hop Question Answering under Temporal Knowledge Editing

**Keyuan Cheng**[*,1,2,3], **Gang Lin**[*,1,2,3], **Haoyang Fei**[*,1,2,3], **Yuxuan zhai**[3],
**Lu Yu**[5], **Muhammad Asif Ali**[1,2], **Lijie Hu**[†,1,2,4], **and Di Wang**[†,1,2,4]
[1]Provable Responsible AI and Data Analytics (PRADA) Lab
[2]King Abdullah University of Science and Technology
[3]South China University of Technology    [4]SDAIA-KAUST AI    [5]Ant Group

## Abstract

Multi-hop question answering (MQA) under knowledge editing (KE) has garnered significant attention in the era of large language models. However, existing models for MQA under KE exhibit poor performance when dealing with questions containing explicit temporal contexts. To address this limitation, we propose a novel framework, namely **TEMP**oral know**LE**dge augmented **M**ulti-hop **Q**uestion **A**nswering (Temple-MQA). Unlike previous methods, Temple-MQA first constructs a time-aware graph (TAG) to store edit knowledge in a structured manner. Then, through our proposed inference path, structural retrieval, and joint reasoning stages, Temple-MQA effectively discerns temporal contexts within the question query. Experiments on benchmark datasets demonstrate that Temple-MQA significantly outperforms baseline models. Additionally, we contribute a new dataset, namely TKEMQA, which serves as the inaugural benchmark tailored specifically for MQA with temporal scopes.

## 1   Introduction

Large Language Models (LLMs) have garnered widespread attention owing to their remarkable capacity for knowledge comprehension, enabling tailored solutions across various applications (Zhao et al., 2023; Huang & Chang, 2022). However, the presence of outdated knowledge presents a significant challenge, impeding the ability of LLMs to provide accurate responses regarding recent events and facts – a phenomenon known as hallucinations, wherein LLMs tend to fabricate plausible yet incorrect responses about unknown facts (Hong et al., 2023). Thus, ensuring the timely updating of LLMs with the latest information is of paramount importance. However, as the most direct approach for updating information, editing LLMs by re-training from scratch is practically infeasible, as it requires huge computational resources and substantial investments. In response, *Knowledge Editing* (KE) has emerged as a focal point, which aims to precisely modify or update the knowledge in LLMs without requiring model retraining. This topic has garnered considerable attention in recent years (Wang et al., 2023b; Zhang et al., 2024).

Multi-hop question answering (MQA), on the other hand, aims to tackle complex inquiries that necessitate multiple reasoning steps. In the context of MQA, KE introduces the concept of *"ripple effects"*, wherein a single edit may trigger a cascade of subsequent knowledge edits or updates (Cohen et al., 2023). For instance, if we update the knowledge about the U.S. president from *Trump* to *Biden*, correspondingly knowledge for the question: "Who is the wife of the U.S. president?" should also be updated (Cohen et al., 2023). There are two prominent lines of work in this area: parameter-based editing and memory-based editing. Parameter-based editing methods update the knowledge by directly modifying the parameters of the model (Meng et al., 2022a;b), while memory-based methods employ an explicit memory to store information about the facts to be modified (Mitchell et al., 2022;

---

*The first three authors contributed equally to this work.
†Correspondence to Lijie Hu {lijie.hu@kaust.edu.sa} and Di Wang {di.wang@kaust.edu.sa}.

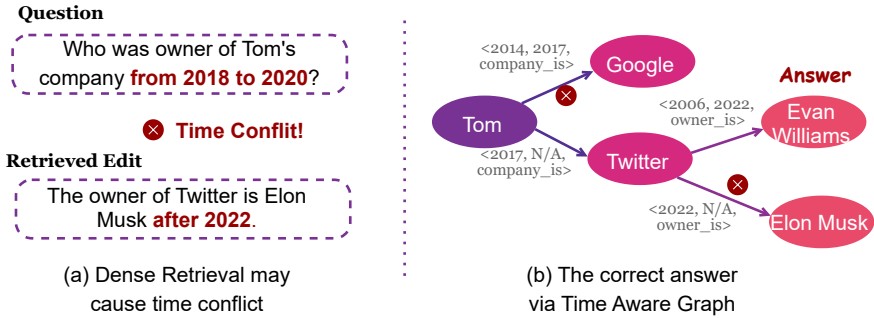

**Question**

Who was owner of Tom's company **from 2018 to 2020**?

❌ **Time Conflit!**

**Retrieved Edit**

The owner of Twitter is Elon Musk **after 2022**.

(a) Dense Retrieval may cause time conflict

(b) The correct answer via Time Aware Graph

Figure 1: **Limitations of dense retrieval to handle temporal context.** (a) There is a temporal constraint in question. However, the dense retrieval mechanism fails to accurately capture this temporal information, thus yielding an erroneous retrieval. (b) Our time-aware graph stores edits as a time-conscious knowledge structure to help distinguish temporal context and improve retrieval accuracy.

Zhong et al., 2023; Gu et al., 2023). In most cases, the memory-based approaches outperform the parameter-based methods (Gu et al., 2023; Zhong et al., 2023)

To address ripple effects, memory-based methods adopt a plan-and-solve paradigm (Khot et al., 2022; Wang et al., 2023a), wherein LLMs are prompted to decompose multi-hop questions into sub-questions, followed by iteratively answering each sub-question. These methods employ dense retrieval to identify relevant edits for each sub-question by comparing semantic similarities (Karpukhin et al., 2020). However, we observed that these approaches exhibit a key limitation, i.e., they perform poorly on questions with explicit temporal contexts (Yin et al., 2023). This inability to cater to temporal context is attributed to the dense retrieval employed by the memory-based methods, which primarily store the edit retrieval information in an unstructured format. This is illustrated in Figure 1 (a), where the dense retrieval retrieves an irrelevant fact (from the year 2022) for the question "Who was the owner of Tom's company in 2020?".

To address the aforementioned limitations, we propose a method called TEMPLE-MQA: **TEMP**oral know**LE**dge augmented **M**ulti-hop **Q**uestion **A**nswering. TEMPLE-MQA first constructs a time-aware graph (TAG) to store edit information in a structured format to effectively preserve context-specific temporal information in the best possible way (Figure 1 (b)). To enhance retrieval performance across semantically related edits, TEMPLE-MQA: (i) leveraging data augmentation techniques to capture aliases for entity names, aiding in entity disambiguation, and (ii) employing context-dependent concepts to explicitly filter edits based on contextual similarity. To tackle multi-hop questions, TEMPLE-MQA utilizes pre-trained LLMs to devise an inference path and conducts step-by-step joint reasoning, leveraging both LLMs and TAG to derive the final response. Experimental evaluation using benchmark data sets shows that TEMPLE-MQA outperforms the existing state-of-the-art approaches on MQA by a significant margin. We summarize the key contributions of this work as follows.

- We propose TEMPLE-MQA, the first method capable of achieving high accuracy in MQA under a substantial volume of temporal knowledge edits without forgetting historical knowledge.

- Unlike previous approaches, TEMPLE-MQA constructs a TAG to store structured information through our devised methods. Additionally, we propose a novel planning procedure and a joint reasoning approach for the inference path, alongside the development of a unique structural retrieval procedure tailored for knowledge retrieval, with potential applicability to other problems.

- Extensive experimental results on two benchmark datasets show that TEMPLE-MQA outperforms the previous seven baselines via different metrics for MQA under massive edits. Furthermore, we develop a new dataset, namely TKEMQA, which serves as the first benchmark on MQA with temporal scopes.

## 2 Related Work

**Parameter-based Editing.** Parameter-editing approaches aim to update a model by incorporating information about updated data or knowledge while ensuring minimal changes to predictions on other data points. These approaches can be categorized into fine-tuning, locating and editing, and meta-learning methods. Fine-tuning methods utilize new knowledge to fine-tune the model parameters while at the same time combating catastrophic forgetting (Chen et al., 2020; Zhu et al., 2020). Locate and edit approaches treat the layers of a feed-forward network as primary knowledge storage units and update their parameters to edit knowledge. Examples include ROME (Meng et al., 2022a) and its extended version, MEMIT (Meng et al., 2022b), which targets a large number of edits. Huang et al. (2024) investigated the generalization aspects of knowledge edits, while Li et al. (2023) proposed PMET for precise updating of FFN weights. Hu et al. (2024) proposed a wise-layer KE method to facilitate lifelong editing. METO (Yin et al., 2023) employs time information as an optimization goal to learn new knowledge without forgetting historical information. Meta-learning approaches treat the editing task as a machine learning challenge, with examples including hyper-networks trained with constrained optimization for fact-only modification (Cao et al., 2021), belief-graph by Hase et al. (2021), and context-aware meta-learned loss scaling by Hu et al. (2023). However, these methods often underperform for MQA under KE, primarily because the update in the model parameters is more effective for single-hop settings and hard to adapt for multi-hop questions requiring complex reasoning. In contrast, TEMPLE-MQA stores the edits in an edit memory alongside a reasoning link, enabling the decomposition and step-by-step solution of multi-hop questions.

**Memory-based Editing.** These techniques store edits in explicit memory and utilize retrieval-augmented methods to reason over relevant edits and modulate end predictions of the model. For instance, SERAC (Mitchell et al., 2022) introduces semi-parametric editing coupled with a retrieval-augmented counterfactual model. GRACE (Hartvigsen et al., 2022) embeds additional adapters within LLMs for editing, using vector matching to locate and modify edited knowledge. IKE (Zheng et al., 2023) employs in-context learning based on demonstration storage to edit the model's knowledge. MeLLo (Zhong et al., 2023) is a simple memory-based approach that stores all edited facts externally and prompts the language model during inference. Additionally, Zhong et al. (2023) also introduces the MQUAKE benchmark for evaluating the MQA performance of their model. Recently, PokeMQA (Gu et al., 2023) proposes a two-stage process of decoupling self-checking and sub-question decomposition to enhance retrieval performance. DeepEdit (Wang et al., 2024) utilizes a depth-first constrained decoding method to edit knowledge for MQA. Tilp (Xiong et al., 2023) provides a differentiable method for learning temporal knowledge graphs in KGs. TEILP (Xiong et al., 2024b) proposes a logical reasoning framework for time prediction in temporal knowledge graphs. TG-LLM (Xiong et al., 2024a) proposes a framework that enhances temporal reasoning capabilities in large language models. Note that, unlike existing solutions, TEMPLE-MQA does not require sub-problem decomposition but generates an inference path directly, which is more straightforward and can directly generate a complete plan. Moreover, the structural retrieval method proposed by our TEMPLE-MQA does not require fine-tuning compared to PokeMQA, making it applicable to a broader range of scenarios, including the processing of temporal information.

## 3 Preliminaries

**Notations.** We represent the set of facts as $\mathcal{D} = \{(s, r, o)\} \subseteq \mathcal{E} \times \mathcal{R} \times \mathcal{E}$, where $\mathcal{E}, \mathcal{R}$ denote the set of entities and relations respectively. A tuple $(s, r, o) \in \mathcal{D}$ represents a fact, with subject entity $s$ and object entity $o$ having relation $r$. For the temporal facts, we add time information to $\mathcal{D}$, i.e., $\mathcal{D}_t = \{(s, r, o, \tau_s, \tau_e)\} \subseteq \mathcal{E} \times \mathcal{R} \times \mathcal{E} \times \mathcal{T} \times \mathcal{T}$, where $\mathcal{T}$ represents the timestamps. A tuple $(s, r, o, \tau_s, \tau_e) \in \mathcal{D}_t$ represents a temporal fact, with subject entity $s$, object entity $o$, having relation $r$ with temporal scope from $\tau_s$ to $\tau_e$. We use $\mathcal{G}_t$ to represent the time aware graph constructed using $\mathcal{D}_t$ (Section 4.1).

### 3.1 Knowledge Editing and MQA

**Knowledge Editing (KE).** A KE request can be represented as $\mathcal{F} = \{f_1, f_2, \cdots f_n\}$, which contains a set of fact edits. Each edit $f_i \in \mathcal{F}$ represents an individual knowledge editing

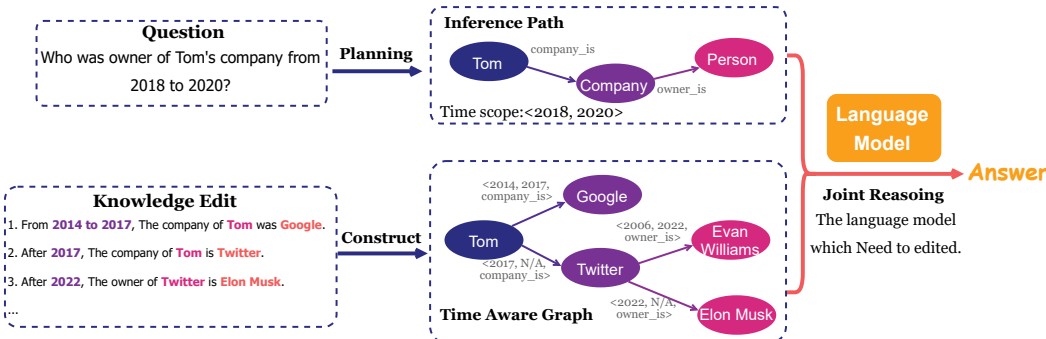

Figure 2: **Overview of TEMPLE-MQA**. Given a multi-hop question, TEMPLE-MQA employs LLMs to strategize an inference path for the question. Then, it leverages LLMs and the TAG ($\mathcal{G}_t$) for joint reasoning on the inference path.

operation, denoted by $f_i = (s, r, o) \rightarrow (s, r, o^*)$, indicating that the object of the subject $s$ with relation $r$ is updated from $o$ to $o^*$.

**MQA under KE.** A multi-hop question $Q$ requires multiple reasoning steps to derive the final answer. The reasoning steps could be represented as a *chain of facts* $\langle (s_1, r_1, o_1), \cdots, (s_n, r_n, o_n) \rangle$, where each step (hop) in the chain $(s_i, r_i, o_i)$ associated with an individual fact. The subject and object are chained together, i.e., the object $(o_i)$ from the preceding fact becomes the subject $(s_{i+1})$ for the subsequent fact.

If one of the steps $(s_i, r_i, o_i)$ in $Q$ is associated with a fact edit $f_i \in \mathcal{F}$, it causes ripple effects, i.e., all subsequent facts have to be updated in order to derive the final answer to the question. Mathematically, a chain of facts in $Q$ coupled with corresponding fact edits may be represented as: $\langle (s_1, r_1, o_1), \cdots, (s_i, r_i, o_i^*), \cdots, (s_n^*, r_n, o_n^*) \rangle$. Note that $Q$ may initiate multiple fact edits in $\mathcal{F}$. The end goal of MQA under KE is to derive the final answer $o_n^*$ for $Q$ after considering all associated edits in $\mathcal{F}$.

### 3.2 Temporal Scope

**Temporal KE.** In KE under temporal scope, we use $\mathcal{F}_t = \{f_1, \cdots, f_n\}$ to represent the temporal fact edits, with a single temporal fact edit denoted by $f_i = (s, r, o, \tau_s, \tau_e) \rightarrow (s, r, o^*, \tau_s^*, \tau_e^*)$. It means that $(s, r, o)$ is valid from $\tau_s$ until $\tau_e$, and after that $(s, r, o^*)$ is valid from $\tau_s^*$ to $\tau_e^*$.

**Temporal MQA under KE.** Similar to MQA under KE, here, each reasoning step will be represented as a temporal fact. A chain of facts in $Q$ coupled with corresponding temporal fact: $\langle (s_1, r_1, o_1, \tau_{1,s}, \tau_{1,e}), \cdots, (s_i, r_i, o_i^*, \tau_{1,s}^*, \tau_{1,e}^*), \cdots, (s_n^*, r_n, o_n^*, \tau_{n,s}^*, \tau_{n,e}^*) \rangle$.

## 4 TEMPLE-MQA

TEMPLE-MQA is a generalized memory-based editing method. It is capable of MQA under knowledge editing (including knowledge correction and updating, etc.) without forgetting historical knowledge. As shown in Figure 2, the core components of TEMPLE-MQA include pre-existing LLMs and a Time Aware Graph (TAG). The workflow of TEMPLE-MQA is summarized as follows: (i) Constructing a TAG for the given edits; (ii) Utilizing LLMs to devise an inference path for each multi-hop question; (iii) Finally, employing the LLMs along with TAG via our proposed structural retrieval mechanism for joint reasoning based on the inference path. We will discuss each component in detail.

### 4.1 Time-Aware Graph Construction

The objective of the graph construction process is to design a mechanism to store knowledge edits in a structured format, facilitating effective and efficient retrieval. It is worth noting that this approach markedly differs from existing approaches, which predominantly store knowledge edits in an unstructured format (Zhong et al., 2023). For example, "The president of the U.S. is Joe Biden." is an unstructured text format, while (U.S., president is, Joe Biden) is

structured. This distinction proves crucial in overcoming the challenge of handling temporal contexts for KE.

Historical and current facts are typically stored in unstructured text form as $f_{\text{old}}$ and $f_{\text{cur}}$. The graph construction process initially transforms them into structured form $\mathcal{F}_t$ (detailed in Section 3.2). Then it utilizes these structured temporal edits $\mathcal{F}_t$ as input and generates a graph $\mathcal{G}_t$ as output. The process flow is elucidated as follows.

**Converting unstructured facts to a structured form.** First, we employ a pre-trained LLM to convert all unstructured facts $f_x$ (include $f_{\text{old}}$ and $f_{\text{cur}}$) into a structured form to explicate the information content across different dimensions. To achieve this, we design an in-context learning prompt (see Appendix A for details) that prompts an LLM to process the unstructured fact.

$$s, r, o, \tau_s, \tau_e, c(o) = \text{LLM}(\text{P}_{convert}(f_x)), \tag{1}$$

Where $f_x$ denotes the unstructured fact, LLM yields a structured output $(s, r, o, \tau_s, \tau_e, c(o))$. Here, $c(o)$ represents the concept associated with the object entity $o$ Ali et al. (2020), and $\text{P}_{convert}$ is the in-context learning prompt. We use GPT-3.5-turbo-instruct as the pre-trained LLM. Finally, for all pairs of historical and current facts $(f_{\text{old}}, f_{\text{cur}})$, via our above procedure, we can get the set of structured temporal fact edits $\mathcal{F}_t = \{f_x \rightarrow (s, r, o, \tau_s, \tau_e)\}$.

It is notable that compared to the original temporal fact, here we also include the concept information for the object, $c(o)$, with the fact structure. It provides multi-faceted benefits, including (i) disambiguation of the entities to their appropriate concepts based on the context and (ii) improved performance for edit retrieval by offering a wide range of fine-grained concepts (Ling & Weld, 2012). For instance, in different contexts, the entity "Washington" can refer to a city name or a person's name, and without including the concept $c(o)$, explicitly storing this information poses a challenge to distinguishing between them.

**Data augmentation.** The purpose of data augmentation is to capture different possible lexical variations for the same entity. This step is pivotal for improving retrieval performance, as it assists in filtering out edits with subjects different from the query subject. This eventually helps reduce the search space by filtering numerous irrelevant edits. We utilize SPARQL (detailed in Appendix E) to get all possible aliases of the subject entity $s$ from Wikidata, denoted as $A(s) = \{s_0, s_1, s_2, \cdots\}$, where $s_0 = s$ and $s_i$ is the $i$-th alias for $s$. Note that the current formulation of TEMPLE-MQA favors data augmentation on the subject dimension.

**Graph construction.** Finally, we utilize the structured information about historical and current knowledge to construct the TAG $\mathcal{G}_t$. Specifically, we store the current knowledge as $\{(n_s : s_i, e : (r, c(o), \tau_s^*, \tau_e^*), n_e : o^*)\}$ for $s_i \in A(s)$, and the historical knowledge as $\{(n_s : s_i, e : (r, c(o), \tau_s, \tau_e), n_e : o)\}$ for $s_i \in A(s)$. Here $n_s$, $e$, and $n_o$ represent the start node, edge, and end node in $\mathcal{G}_t$, respectively.

We can also use the above steps to construct graphs for non-temporal edits with the distinction that we exclude temporal scopes. In this scenario, we only store the updated knowledge as $\{(n_s : s_i, e : (r, c(o)), n_e : o^*)\}$ for $s_i \in A(s)$. Note, for $\mathcal{G}$, we only store current knowledge.

## 4.2 Planning and Reasoning on Inference Path

In this section, we introduce the planning and reasoning stages of TEMPLE-MQA: i) The planning stage leverages LLMs to generate an inference path $P$ and extract temporal scope for the question $Q$. ii) The reasoning stage performs a step-by-step joint reasoning by utilizing LLMs and the TAG we constructed in Section 4.1 to solve the question $Q$ using the inference path $P$. We will discuss them in detail.

**Stage 1: Inference Path.** This stage aims to exploit the instruction-following ability of LLMs to generate a structural inference path $P$ that is helpful for answering $Q$. To achieve this, we design an in-context learning prompt (explained in Appendix A) that prompts the LLMs to generate an inference path along with the temporal scopes, as shown below.

$$P, \tau_s^Q, \tau_e^Q = \text{LLM}(P_{\text{infer}}(Q)), \tag{2}$$

Figure 3: **The workflow of structural retrieval.** 1) The first step is to extract the 1-hop sub-graph where the center point is Tom. We filter out the edits not conforming with the temporal scope of the query. 2) The second step calculates the semantic similarity of the relation and concept between the query and knowledeg in sub-graph.

where $P_{\text{infer}}$ is the in-context prompt for $Q$ used as input for LLM, yielding the inference path $P$ and temporal scopes $\tau_s^Q$ and $\tau_e^Q$ as outputs. $P$ is represented as $\langle (s_1, r_1, c(o_1)), \cdots, (c(o_{n-1}), r_n, c(o_n)) \rangle$, where $s_1$ is an entity extracted from question $Q$.

Note that, unlike previous research, our approach is more practical and efficient, as it computes the inference path in one go without needing to alternate between plan and solve phases. The inference path has the concept of entity added among the relation compared to the relation path Luo et al. (2023), which helps us in retrieval.

**Stage 2: Joint Reasoning.** As an iterative process, this stage aims to obtain the final answer for $Q$ by reasoning on $P$. Consider the $i$-th step ($1 \leq i \leq n$), we utilize $(s_i, r_i, c(o_i), \tau_s^Q, \tau_e^Q)$ as input to infer the specific entity $o_i$, where $s_i$ is the output of last step $o_{i-1}(i > 1)$ or start point $s_1$ in $P$, $r_i$ and $c(o_i)$ originate from $P$. For this, TEMPLE-MQA uses the structural retrieval $\mathbf{R}_{\text{struct}}$ (Section 4.3) to retrieve relevant knowledge from the $\mathcal{G}_t$ for the query.

$$o_R^*, \alpha_R = \mathbf{R}_{\text{struct}}(s_i, r_i, c(o_i), \tau_s^Q, \tau_e^Q), \tag{3}$$

where $o_R^*$ is the possible updated knowledge about $o_i$, and $\alpha_R$ denotes the similarity between updated knowledge and query. Note that in our approach, there is no need to use self-checking Zhong et al. (2023) to check if the retrieved fact contradicts the generated answer. Underlying reason for this is the fact that many sota LLMs are trained using RLHF, due to which they exhibit a higher tendency to reject external knowledge that contradicts their own knowledge.

Finally, based on the similarity score $\alpha_R$ compared against a threshold ($\theta$), we decide if we may use retrieved knowledge as such or use the language model for response generation:

$$o_i = \begin{cases} o_R^* & \alpha_R > \theta \\ \mathbf{M}_{\text{query}}(P_{\text{query}}(s_i, r_i, c(o_i), \tau_s, \tau_e)) & \alpha_R \leq \theta, \end{cases} \tag{4}$$

where $\mathbf{M}_{\text{query}}$ is an LLM , and $P_{\text{query}}$ is the prompt for query (details are given in Appendix A). We re-iterate the above-mentioned process until we completely exhaust the inference path $P$, yielding $o_n$ as the final answer.

### 4.3 Structural Retrieval

Given a query $(s_i, r_i, c(o_i), \tau_s^Q, \tau_e^Q)$, this section introduces how to retrieve the knowledge from $\mathcal{G}_t$ using structural retrieval $\mathbf{R}_{\text{struct}}$. The workflow of $\mathbf{R}_{\text{struct}}$ (See Figure 3) can be represented by the following steps: (i) Extracting a subgraph, i.e., filtering out the knowledge that does not meet the required subject and temporal scope. (ii) Re-ranking, i.e., re-ranking the remaining knowledge based on semantic similarity of relation and concept with query.

**Step 1: Extracting a Subgraph.** This step extracts a 1-hop subgraph $\mathcal{G}_{sub}$ from $\mathcal{G}_t$ by filtering out relation pairs violating the subject and temporal scopes. Formally, this is represented as:

$$\mathcal{G}_{sub} = \{(n_s, e, n_e) \in \mathcal{G}_t \mid (n_s == s_i) \wedge (e[\tau_s] \geq \tau_s^Q) \wedge (e[\tau_e] \leq \tau_e^Q)\}, \tag{5}$$

where $n_s$ indicates the subject and $(e[\tau_s], e[\tau_e])$ represent the temporal scope of the fact.

**Step 2: Re-ranking.** This step aims to re-rank the candidate answers and find the one with the highest similarity. For this, we first use an existing encoder $E$ to encode the relation and concept for the query, as follows:

$$v_r^q = E(r_i); v_{c(o)}^q = E(c(o_i)). \tag{6}$$

Finally, we re-rank knowledge in $\mathcal{G}_{sub}$ based on the cosine similarity (*sim*) for both relation and fine-grained concept between knowledge and query and then select the knowledge with the highest similarity as the retrieved result.

$$o_R^*, \alpha_R = \underset{(n_s, e, n_e) \in \mathcal{G}_{sub}}{\arg\max} \; (sim(E(e[r]), v_r^q) + sim(E(e[c(o)]), v_{c(o)}^q)). \tag{7}$$

Where $o_R^*$ is the retrieved object, and $\alpha_R$ is the corresponding similarity returned by $\mathbf{R}_{struct}$, $e[r]$ and $e[c(o)]$ indicate relation and concept of the knowledge.

# 5 Experiments

Here we conduct practical evaluations for TEMPLE-MQA compared against different baseline models. Additional results, including the ablation study, can be found in Appendix C.

## 5.1 Experimental Settings

**Datasets.** We evaluate TEMPLE-MQA on a blend of publicly available benchmarks and self-curated datasets. These include: MQUAKE (Zhong et al., 2023), ATOKE (Yin et al., 2023), and our newly proposed dataset TKEMQA. Detailed descriptions and statistics of the datasets are given in Appendix B.1 and D.

**Baselines.** We compare the performance of TEMPLE-MQA against a wide range of parameter-based and memory-based KE methods. The parameter-based baselines include: Fine-tuning (FT) Zhu et al. (2020), ROME (Meng et al., 2022a), MEMIT (Meng et al., 2022b), MEND Mitchell et al. (2021) and METO (Yin et al., 2023). The memory-based baselines include: MeLLo (Zhong et al., 2023), and PokeMQA (Gu et al., 2023). Details are given in Appendix B.2.

**Evaluation metrics.** Similar to the baseline models, we use five different evaluation metrics: (i) Multi-hop Accuracy (M-Acc) (Zhong et al., 2023), (ii) Hop-wise Accuracy (H-Acc) (Gu et al., 2023), (iii) Historical Explicit time Question Score (HES) (Yin et al., 2023), (iv) Current Explicit time Question Score (CES) (Yin et al., 2023), (v) Historical Explicit time Question Score (CES-P) (Yin et al., 2023), (vi) Current Relative time Question Score (CRS) (Yin et al., 2023), and (vii) Historical Relative time Question Score (HRS) (Yin et al., 2023). For all metrics, a larger value indicates that the method is better. Details of evaluation metrics are given in Appendix B.3.

**Experimental setup.** To evaluate performance with varying numbers of edits, we follow the setting of PokeMQA (Gu et al., 2023) to conduct stratified sampling (Parsons, 2014) of the dataset based on the number of hops in questions. This allows us to construct edit batches of different sizes while ensuring a relatively consistent proportion of questions with different hop counts within each batch. We inject all the edits within a batch simultaneously (Zhong et al., 2023). The scenarios of different batch sizes are denoted as M-edited ($M \in \{1, 100, All\}$). We use LLaMa-2-7B (Touvron et al., 2023), Vicuna-7B (Chiang et al., 2023), GPT-turbo-3.5-instruct and GPT-J-6B Wang & Komatsuzaki (2021) as the LLMs and utilize `all-MiniLM-L12-v2`[1] as the encoder in our method. We conducted each experiment four times and reported the average values in the table. All experiments are performed using PyTorch 2.1.2 and RTX 4090 24GB GPU.

## 5.2 Experimental Results

**Non-temporal MQA under KE.** We initially test the performance of TEMPLE-MQA for non-temporal MQA, whose results for MQUAKE are shown in Table 1. These results clearly

---

[1]`https://huggingface.co/sentence-transformers/all-MiniLM-L12-v2`

| Method | MQUAKE-CF-3K | | | | | | MQUAKE-T | | | |
| | 1-edited | | 100-edited | | All-edited | | 1-edited | | All-edited | |
| | M-Acc | H-Acc | M-Acc | H-Acc | M-Acc | H-Acc | M-Acc | H-Acc | M-Acc | H-Acc |
|---|---|---|---|---|---|---|---|---|---|---|
| LLAMA-2 | | | | | | | | | | |
| FT* | 28.20 | 7.3 | 2.37 | 0.03 | - | - | 56.48 | 33.89 | 1.02 | 0.37 |
| ROME* | 13.13 | 5.37 | 3.50 | 0.03 | 3.63 | 0.1 | 24.89 | 17.99 | 1.71 | 0.32 |
| MEMIT* | 14.97 | 6.43 | 9.40 | 2.47 | 2.30 | 0.37 | 30.89 | 23.98 | 25.21 | 20.13 |
| MeLLo | 33.57 | 9.9 | 20.00 | 10.07 | 17.33 | 9.9 | 65.78 | 55.27 | 57.69 | 44.55 |
| PoKeMQA* | 44.13 | 30.6 | 37.33 | 27.83 | 32.83 | 23.87 | 75.43 | 60.44 | 74.36 | 60.22 |
| TEMPLE-MQA (Ours) | **68.32** | **59.46** | **48.95** | **35.17** | **42.20** | **27.51** | **77.56** | **64.87** | **75.73** | **62.30** |
| VICUNA-7B | | | | | | | | | | |
| MeLLo* | 30.70 | 20.84 | 24.75 | 12.25 | 22.35 | 10.18 | 60.72 | 48.55 | 51.55 | 42.97 |
| PoKeMQA* | 45.83 | 34.83 | 38.77 | 31.23 | 31.63 | 25.3 | 74.57 | 55.19 | 73.07 | 55.09 |
| TEMPLE-MQA (Ours) | **71.61** | **62.75** | **56.65** | **44.26** | **46.60** | **37.33** | **81.77** | **69.46** | **78.29** | **68.15** |
| GPT-3.5-TURBO-INSTRUCT | | | | | | | | | | |
| MeLLo* | 57.43 | 28.8 | 40.87 | 28.13 | 35.27 | 25.3 | 88.12 | 52.84 | 74.57 | 53.53 |
| PoKeMQA* | 67.27 | 56.37 | 56.00 | 49.63 | 48.87 | 39.77 | 78.16 | 68.09 | 76.98 | 67.88 |
| TEMPLE-MQA (Ours) | **78.11** | **63.45** | **67.21** | **55.33** | **53.68** | **40.05** | **90.57** | **81.90** | **82.26** | **74.33** |

Table 1: **Experimental results for MQUAKE.** We **boldface** the best-performing scores with the second-best underlined. The result from the previous paper is marked as *. By default, the same symbols are used in the following tables.

show that TEMPLE-MQA consistently outperforms the baseline models across different settings by a large margin. For example, when consider the MQUAKE-CF-3K dataset and M-Acc as the evaluation metric, TEMPLE-MQA on average improved by 91.2%, 112.7% and 101.4% compared to Mello for $\{1, 100, All\}$-edited respectively across three LLMs, and by 42.6%, 32.4%, and 28.6% compared to PokeMQA. We attribute such drastic performance improvement to the following factors: (a) The inference path employed by TEMPLE-MQA is more reliable, which helps the model to boost the end performance significantly; (b) The structural retrieval pipeline (i.e., TAG) designed for TEMPLE-MQA is robust to reduce the number of false positives significantly. We provide an analysis of these factors in Section 5.3.

Compared TEMPLE-MQA with different LLMs, we can see it exhibits superior performance on GPT-3.5-turbo-instruct followed by Vicuna-7B. This is due to the fact that GPT-3.5-turbo-instruct has the strongest reasoning capabilities Chiang et al. (2023), enabling our model to generate more reliable inference paths. Another noteworthy observation in Table 1 is that parameter-based approaches perform poorly compared to memory-based ones, which is consistent with the results in (Zhong et al., 2023).

**Single-hop MQA under Temporal KE.** We then consider single-hop question answering with temporal knowledge, whose results for ATOKE are shown in Table 3. Compared to Table 1, the results show varying behavior, with parameter-based approaches outperforming memory-based methods in some cases, which is contributed by their stronger ability to retain updated knowledge. Specifically, we can see that parameter-based approaches (such as ROME and MEMIT) can achieve very good performance for CES and CES-P, indicating that they can effectively inject new knowledge into the model. However, they perform poorly on the HES metric: even after using METO, the average HES is only 30.5%. This indicates that these methods forget historical knowledge after editing.

On the other hand, MeLLo and PokeMQA achieved higher HES scores but lower CES scores, indicating they are unable to distinguish historical knowledge and edited knowledge. TEMPLE-MQA can take into account both historical and new knowledge. It not only has higher HES than MeLLo and PokeMQA but also maintains high CES. Specifically, on ATOKE-SE, TEMPLE-MQA achieves HES of 97.25, an improvement of 221% compared to MEMIT$_{METO}$. The reason for such a big improvement is that TEMPLE-MQA can effectively distinguish different knowledge in the same time chain through TAG.

**MQA under Temporal KE.** We finally consider the performance on TKEMQA, which consist of $\{1, 2, 3, 4\}$-hops equation answering with temporal knowledge. The results are shown in Table 2. We can see that similar to the above results, TEMPLE-MQA achieves good performance for M-Acc. However, TEMPLE-MQA is worse than MeLLo for HES in the All-edited setting using GPT. This is because there is no historical knowledge matching the question in the edit memory in such a setting. MeLLo's utilization of a self-checking method effectively validates the retrieved results, contributing to its superiority.

| Method | TKeMQA | | | | | | TKeMQA-HK | | | | | |
|---|---|---|---|---|---|---|---|---|---|---|---|---|
| | 1-edited | | 100-edited | | All-edited | | 1-edited | | 100-edited | | All-edited | |
| | M-Acc | HES | M-Acc | HES | M-Acc | HES | M-Acc | HES | M-Acc | HES | M-Acc | HES |
| VICUNA | | | | | | | | | | | | |
| MeLLo | 39.02 | **36.54** | 31.71 | **35.56** | 25.51 | 32.21 | 25.76 | 62.76 | 18.53 | 60.26 | 15.21 | 56.89 |
| PoKeMQA | 36.65 | 33.12 | 30.15 | 31.90 | 23.02 | 29.56 | 22.13 | 57.83 | 20.35 | 56.71 | 16.94 | 52.61 |
| TEMPLE-MQA (Ours) | **52.18** | 35.65 | **50.41** | 34.62 | **48.13** | **34.51** | **51.92** | **77.51** | **50.13** | **75.59** | **48.20** | **75.52** |
| GPT-3.5-TURBO-INSTRUCT | | | | | | | | | | | | |
| MeLLo | 82.14 | 37.75 | 59.17 | 37.19 | 47.53 | **36.57** | 56.04 | 73.44 | 38.22 | 72.63 | 31.45 | 70.75 |
| PoKeMQA | 70.18 | 27.85 | 53.54 | 24.55 | 43.80 | 20.26 | 50.27 | 67.91 | 33.83 | 64.37 | 27.42 | 60.65 |
| TEMPLE-MQA (Ours) | **85.53** | **41.26** | **83.47** | **38.75** | **80.50** | 33.11 | **85.51** | **82.28** | **84.12** | **82.22** | **80.09** | **81.61** |

Table 2: **Experimental results on TKeMQA.**

| Method | AToKe-SE | | | AToKe-ME | | | AToKe-EE | |
|---|---|---|---|---|---|---|---|---|
| | CES | CES-P | HES | CES | CES-P | HES | CES | CES-P |
| GPT-J-6B & 1-EDITED | | | | | | | | |
| FT[*] | 5.73 | 5.69 | 0.06 | 1.11 | 1.18 | 0.03 | 3.41 | 2.91 |
| MEND[*] | 80.47 | 40.56 | 1.73 | 71.83 | 27.96 | 0.40 | 91.94 | 62.48 |
| ROME[*] | **99.99** | **97.01** | 2.41 | 98.85 | 91.54 | 0.44 | 99.93 | 98.70 |
| MEMIT[*] | 99.66 | 92.23 | 2.22 | 98.42 | 91.06 | 0.48 | 99.92 | 95.82 |
| FT$_{METO}$[*] | 2.8 | 2.62 | 3.38 | 1.27 | 1.2 | 1.64 | - | - |
| MEND$_{METO}$[*] | 83.26 | 33.45 | 30.14 | 70.52 | 28.41 | 28.65 | - | - |
| ROME$_{METO}$[*] | 99.95 | 93.78 | 20.25 | **99.93** | 90.97 | 23.22 | - | - |
| MEMIT$_{METO}$[*] | 86.4 | 85.32 | 30.31 | 92.73 | 85.75 | 36.2 | - | - |
| GPT-J-6B & ALL-EDITED | | | | | | | | |
| Mello | 83.78 | 81.55 | 48.19 | 60.82 | 59.15 | 25.65 | **99.97** | 98.60 |
| PokeMQA | 90.91 | 87.66 | 62.49 | 72.73 | 70.92 | 40.99 | 99.87 | 98.62 |
| TEMPLE-MQA (Ours) | **97.95** | 95.88 | **97.25** | 96.46 | **95.62** | **96.43** | 99.92 | **98.76** |

Table 3: **Experiment results on AToKe.** METO is a plug-and-play method to strengthen the memory of historical knowledge. We mark the METO untested results as "-". AToKe-ME indicates each data includes two edits to perform continuous knowledge updating.

Moreover, in contrast to the previous two datasets, we find that TEMPLE-MQA on TKeMQA experiences only a slight performance decrease when the edit batch size increases. We further conduct an experiment to study the performance w.r.t. difference edit numbers (see Figure 4 in Appendix). It can be observed from Figure 4 (a) and Figure 4 (b) that as the number of edits increases, the performance of TEMPLE-MQA drops rapidly on the MQuAKE-CF-3K. We find that this is because MQuAKE-CF-3K has unpassable data in the mass-edit setting. Specifically, there are data that are affected by the editing of other data. Under M-edited ($M > 1$) settings, if they are in the same batch, unpassable data will be generated (see Appendix Table 15 for an example). To verify this, we filter out conflicting data and create a new data MQuAKE-CF-3K-FIX. The result is shown in Figure 4 (c). After repairing, we can observe that TEMPLE-MQA is more stable than other methods, which illustrates the accuracy of structural retrieval.

## 5.3 Abalation Study

We first conduct an ablation study on different modules of the structural retrieval: without filtering sub-graphs according to the subject entity (i.e. - w/o Subject), without considering the similarity of relation in the Re-ranking step (i.e. - w/o Relation), without considering the similarity of the concept of object entity (i.e. - w/o Concept).

The results are shown in Table 10 in Appendix. First, we investigate the role of filtering sub-graphs. We can see without filtering out the sub-graph through the subject, TEMPLE-MQA decreased by an average of 8.6% and 7.8% on M-Acc and H-Acc. This is the most impactful component among all removals. Second, we can see both relation and concept can improve the performance of TEMPLE-MQA in all the settings, indicating its necessity. Third, the impact of all eliminated components will become larger as the edit memory increases.

Subsequently, we conduct an ablation study on inference path planning module, with details as follows: (1) Replacing inference path planning with sub-question decomposition by Zhong et al. (2023) (i.e., - w/o inference path). (2) Exploring the influence of $k$-shot ($k \in \{1, 2, 4\}$) on in-context learning prompt.

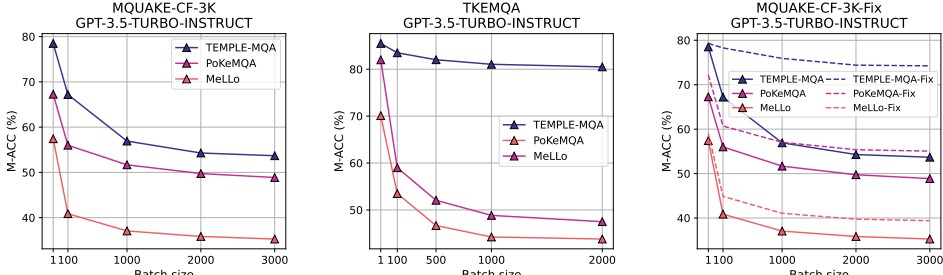

Figure 4: The line graph depicting the decrease in M-Acc with the increase in edit batch size on datasets MQUAKE-CF-3K, TKEMQA and MQUAKE-CF-3K-FIX.

Corresponding results are shown in Table 11 in Appendix. These results show, if we exclude the inference path planing, the M-ACC drops by almost 15.44%, 14.39%, 9.53% respectively. Also, the M-ACC drops about 10.43%, 7.77% and 3.51% for $k$-shot ($k \in \{1, 2, 4\}$) settings. Thus more examples can help LLMs follow instructions better, also the quality and diversity of the examples is important.

## 6  Conclusion

We presented a novel framework TEMPLE-MQA for MQA under temporal knowledge editing. TEMPLE-MQA first constructs a time-aware graph, then utilizes inference paths, structural retrieval, and joint reasoning, capable of enhancing MQA performance while distinguishing the temporal context. We also proposed a benchmark TKEMQA, which is the first benchmark specified for MQA with temporal scopes. Experiments on TKEMQA and the existing two benchmarks demonstrated that TEMPLE-MQA outperforms previous methods.

## 7  Limitations

Our work poses following limitations and we will resolve them in our future works:

- Temple-MQA relies on TAG, which brings additional computational overhead.
- Temple-MQA requires writing prompts manually to adapt to different tasks.
- Although retrieval of Temple-MQA doesn't need fine-tuning an encoder, it has a complex workflow.

**Acknowledgements.**   Di Wang, Lijie Hu and Muhammad Asif Ali are supported in part by the baseline funding BAS/1/1689-01-01, funding from the CRG grand URF/1/4663-01-01, FCC/1/1976-49-01 from CBRC and funding from the AI Initiative REI/1/4811-10-01 of King Abdullah University of Science and Technology (KAUST). Di Wang is also supported by the funding of the SDAIA-KAUST Center of Excellence in Data Science and Artificial Intelligence (SDAIA-KAUST AI). We also acknowledge support from OpenAI API Researcher Access Program.

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

# A   Details of Prompts in TEMPLE-MQA

---

**[Demo]**
Unstructured fact: From 2009 to 2017, Obama was the president of the United States.
Structural form: <United States, the president is, person, 2009, 2017, Obama>
Unstructured fact: From 2019 to 2022, the head of government in kazakhstan is Askar Mamin.
Structural form: <Kazakhstan, head of government, person, 2019, 2022, Askar Mamin>
...Other demo ...
**[Instruction]**
Please refer to the above demo and convert the following edit into the structural form:
<subject, relation, type of object, start time, end time, object>
**[Task]**
<Fact>

---

Table 4: **The prompts $P_{convert}$ to convert unstructured knowledge used in Equation (1)**.
**[Demo]** include several high-quality demonstrations handwritten by humans. <Fact>
indicates the unstructured fact $f_x$. More examples are shown in Table 19.

---

**[Demo]**
Question: Who was the owner of Tom's company from 2018 to 2020?
Inference path: <Tom, company is, company>, <company, owner is, people>
Time: 2018, 2020
Question: What is the religion of the head of government of Israel after 2022?
Inference path: <Israel, head of government, person>, <person, religion is, religion>
Time: 2022, None
...Other demo...
**[Instruction]**
Please refer to the above demo and generate a valid inference path for the following question:
**[Task]**
<Question>

---

Table 5: **The prompt $P_{plan}$ to extract inference path used in Equation (2)**, where <Question>
indicates the question $Q$.

---

**[Demo]**
Inference step: <United Kingdom, head of government, person>
Time: from 2019 to 2022
Answer: Boris Johnson
Inference step: <Christian Wulff, spouse, person>
Time: after 2023
Answer: Bettina Wulff
...Other demo...
**[Instruction]**
Please generate a valid answer like the above example for the following task:
**[Task]**
<Inference step>
<Time>

---

Table 6: **The prompt $P_{query}$ for querying LLM to generate answer used in Equation (4)**,
where <Inference step> indicates specific inference step in the inference path, <Time>
indicates temporal scope.

# B   Experimental Details

## B.1   Dataset

We provide a detailed description of the evaluation datasets as follows.

| Datasets | #Edits | 2-hop | 3-hop | 4-hop | Total |
|---|---|---|---|---|---|
| | 1 | 513 | 356 | 224 | 1093 |
| | 2 | 487 | 334 | 246 | 1067 |
| MQUAKE-CF-3K | 3 | - | 310 | 262 | 572 |
| | 4 | - | - | 268 | 268 |
| | All | 1000 | 1000 | 1000 | 3000 |
| MQUAKE-T | 1 | 1421 | 445 | 2 | 1868 |

Table 7: **Statistics of the MQUAKE dataset**.

| Datasets | ATOKE-SE | ATOKE-ME | ATOKE-EE |
|---|---|---|---|
| Size | 8819 | 8820 | 8819 |

Table 8: **Statistics of the ATOKE dataset**.

**(a) MQUAKE (Zhong et al., 2023).** MQUAKE includes MQUAKE-CF-3K based on counterfactual editing and MQUAKE-T based on real-world changes. These datasets encompass k-hop questions ($k \in \{2, 3, 4\}$), each associated with one or more edits. Statistics are shown in Table 7.

**(b) ATOKE (Yin et al., 2023).** ATOKE is the first temporal knowledge editing dataset containing a series of world knowledge with timestamps, regarded as a series of knowledge updates. ATOKE contains three editing types: single edit (SE), multiple edits (ME), and extending edit (EE), corresponding to three datasets ATOKE-SE, ATOKE-ME and ATOKE-EE. Statistics of these three datasets are shown in Table 8.

- **ATOKE-SE.** Each data of this dataset includes one edit for knowledge updating and a historical question and current question corresponding to the time scope before and after the modification.

- **ATOKE-ME.** Each data of this dataset includes two edits to perform continuous knowledge updating, two historical questions, and one current question.

- **ATOKE-EE.** Each data of this datasets include one edit to modify the time scope of knowledge (not updating the object), and only one current question.

It is worth explaining that the above three datasets are all aimed at single-hop questions only.

**(c) TKEMQA.** TKEMQA is our newly curated benchmark dataset. It is a blend of carefully crafted multi-hop questions with explicit temporal scopes. This data set is designed to evaluate the ability of KE methods rigorously. **TKEMQA-HK** provides additional knowledge corresponding to two historical questions on the basis of TKEMQA, which is used to explore whether the memory-based method will confuse the temporal context. See Section D for a detailed introduction, construction process, and statement temples of TKEMQA.

| Datasets | #UK | #CK | #HK | 1-hop | 2-hop | 3-hop | 4-hop | Total |
|---|---|---|---|---|---|---|---|---|
| | 1 | 0 | 0 | 500 | 334 | - | - | 834 |
| TKEMQA | 1 | 1 | 0 | - | 166 | 500 | 500 | 1166 |
| | | All | | 500 | 500 | 500 | 500 | 2000 |
| | 1 | 0 | 2 | 500 | 334 | - | - | 834 |
| TKEMQA-HK | 1 | 1 | 2 | - | 166 | 500 | 500 | 1166 |
| | | All | | 500 | 500 | 500 | 500 | 2000 |

Table 9: **Statistics of TKEMQA and TKEMQA-HK**. **#UK** represents the number of updated knowledge, **#CK** represents the number of corrective knowledge and **#HK** represents the number of historical knowledge. TKEMQA-HK include two questions asking about historical knowledge which TKEMQA does not include.

## B.2 Baseline Models

**(a) Parameter-based.** The parameter-based baselines include: (i) **Fine-tuning (FT)** (Zhu et al., 2020) that performs a gradient-based update on the model parameters to incorporate new knowledge. (ii) **ROME** (Meng et al., 2022a) first locates factual knowledge at a specific layer of the transformer architecture and then updates the feed-forward network of this layer to insert new knowledge. (iii) **MEMIT** (Meng et al., 2022b) extends ROME to allow modifying a range of feed-forward network layers for a large amount of knowledge. (iv) **MEND** Mitchell et al. (2021) learns a hyper network to produce weight updates by decomposing the gradient of standard fine-tuning into a low-rank form. (v) **METO** (Yin et al., 2023) extends the parameter-based methods by adding the time information as an optimization objective.

**(b) Memory-based.** The memory-based baselines include: (i) **MeLLo** (Zhong et al., 2023) that uses the plan-and-solve paradigm. (ii) **PokeMQA** (Gu et al., 2023) extends MeLLo by adopting a two-stage retrieval process to decouple the question decomposition and knowledge editing.

## B.3 Evaluation Metrics

Details about the evaluation metrics and their mathematical formulation are provided as follows:

(i) **Multi-hop Accuracy (M-Acc)**, which is used to measure the accuracy of the language models on multi-hop questions. For M-Acc, we use the same settings as Zhong et al. (2023). The calculation formula for M-Acc is as follows:

$$\mathbb{1}\left[\bigvee_{q \in \mathcal{Q}} [f^*(q) = a^*]\right].$$ 

(8)

Where $f^*(\cdot)$ represents the edited model, $\mathcal{Q}$ and $a^*$ represent the multi-hop questions and the edited answer for each case, respectively.

(ii) **Hop-wise Accuracy (H-Acc)**, which is used to check the correctness of the intermediate reasoning path for MQA. For H-Acc, we follow the same settings proposed by Gu et al. (2023). Given edited chain of facts $\mathcal{C}^*$, H-Acc is defined as

$$\mathbb{1}\left[\bigwedge_{(s,r,o^*) \in \mathcal{C}^*} [f^*(s,r) = o^*]\right].$$ 

(9)

(iii) **Historical Explicit time Question Score (HES)**, which is the accuracy of explicit temporal questions about historical knowledge (Yin et al., 2023).

(iv) **Current Explicit time Question Score (CES)**, which measures the accuracy of explicit temporal questions about current knowledge (Yin et al., 2023).

(v) **Current Explicit time Paraphrase Question Score (CES-P)**, which is the accuracy of explicit temporal questions of semantic paraphrasing asked about current knowledge (Yin et al., 2023).

HES, CES, and CES-P require the model to be able to recall knowledge at a specific temporal scope, calculated according to the following formula:

$$\mathbb{1}\left[f^*(s, r, \tau_s, \tau_e) = o\right].$$ 

(10)

## C More Experiment Results

**Data Conflict in MQUAKE.** In the analysis for MQA under Temporal KE, we observed that TEMPLE-MQA on TKEMQA experiences only a slight performance decrease when the edit batch size increases, while its decrease is faster for MQUAKE. This is because MQUAKE-CF-3K has unpassable data in the mass-edit setting, i.e., we can observe data

| Method | MQUAKE-CF-3K | | | | | | MQUAKE-T | | | |
| --- | --- | --- | --- | --- | --- | --- | --- | --- | --- | --- |
| | 1-edited | | 100-edited | | All-edited | | 1-edited | | All-edited | |
| | M-Acc | H-Acc | M-Acc | H-Acc | M-Acc | H-Acc | M-Acc | H-Acc | M-Acc | H-Acc |
| - w/o Subject | 77.96 | 62.85 | 62.06 | 51.16 | 44.09 | 34.23 | 89.56 | 80.77 | 73.97 | 66.56 |
| - w/o Relation | 77.78 | 62.67 | 65.15 | 52.88 | 49.53 | 36.51 | 89.72 | 80.80 | 77.13 | 68.49 |
| - w/o Concept | 78.08 | 63.02 | 65.70 | 53.29 | 50.10 | 37.55 | 90.01 | 81.35 | 78.67 | 70.20 |
| TEMPLE-MQA | 78.11 | 63.45 | 67.21 | 55.33 | 53.68 | 40.05 | 90.57 | 81.90 | 82.26 | 74.33 |

Table 10: Ablation study result of TEMPLE-MQA.

| Method | MQUAKE-CF-3K | | |
| --- | --- | --- | --- |
| | 1-edited | 100-edited | All-edited |
| - w/o inference path | 62.67 | 52.82 | 44.33 |
| 1-shot | 65.71 | 55.12 | 46.87 |
| 2-shot | 69.48 | 58.72 | 47.48 |
| 4-shot | 73.93 | 64.63 | 49.91 |
| TEMPLE-MQA(8-shot) | 78.11 | 67.21 | 53.68 |

Table 11: The ablation result of inference path planning on MQUAKE-CF-3K with metric of M-Acc.

| Method | ATOKE | | | |
| --- | --- | --- | --- | --- |
| | CRS(SE) | HRS(SE) | CRS(ME) | HRS(ME) |
| MEND+ | 25.41 | 30.17 | 21.84 | 30.83 |
| ROME+ | 78.88 | 16.29 | 82.40 | 18.18 |
| MEMIT+ | 74.07 | 24.32 | 73.58 | 26.04 |
| TEMPLE-MQA | 85.81 | 78.23 | 87.74 | 80.92 |

Table 12: The results on the AToke-SE and AToke-ME.

| Method | 2-hop | | 3-hop | | 4-hop | |
| --- | --- | --- | --- | --- | --- | --- |
| | Cost($) | Time(s) | Cost($) | Time(s) | Cost($) | Time(s) |
| **Mello** | 0.73 | 8.98 | 0.94 | 12.13 | 1.61 | 16.41 |
| **PokeMQA** | 0.93 | 10.33 | 1.09 | 11.98 | 1.36 | 18.85 |
| **TEMPLE-MQA** | 0.15 | 2.50 | 0.25 | 5.31 | 0.27 | 6.98 |

Table 13: Average expense and time cost comparison of methods on the MQUAKE-CF-3K dataset for solving per multi-hop problem in the same experimental environment.

| Method | 1-edited | | | 100-edited | | | All-edited | | |
| --- | --- | --- | --- | --- | --- | --- | --- | --- | --- |
| | R-Acc | M-Acc | H-Acc | R-Acc | M-Acc | H-Acc | R-Acc | M-Acc | H-Acc |
| **Mello** | 91.6 | 57.43 | 28.8 | 83.1 | 40.87 | 28.13 | 74.5 | 35.27 | 25.3 |
| **PokeMQA** | 95.2 | 67.27 | 56.37 | 89.5 | 56.00 | 49.63 | 81.5 | 48.87 | 39.77 |
| **TEMPLE-MQA** | 99.5 | 78.11 | 63.45 | 93.2 | 67.21 | 55.33 | 87.8 | 53.68 | 40.05 |

Table 14: Retrieval accuracy of different methods on the MQUAKE-CF-3K dataset. R-Acc refers to the accuracy of retrieving edited facts from memory.

conflicts in MQUAKE. These conflicts arise from inconsistent counterfactual edits between two cases containing the same knowledge (i.e., one is edited, but another is not edited). Under mass editing settings, the inference process of one case may be influenced by the editing of another case, leading to incorrect answers. Table 15 provides an example of data conflict in MQUAKE.

**Cost of Inference Path.** Here, we aim to compare the expense and time cost of our inference path to those of previous methods. According to Table 13, we can easily see that, for MQUAKE-CF-3K, TEMPLE-MQA averagely saves 78.7% and 80.4% of the expense compared to Mello and PokeMQA, respectively, and reduces the inference time by 62.9% and 64.8%. The reason is that TEMPLE-MQA only needs to call LLMs once to generate

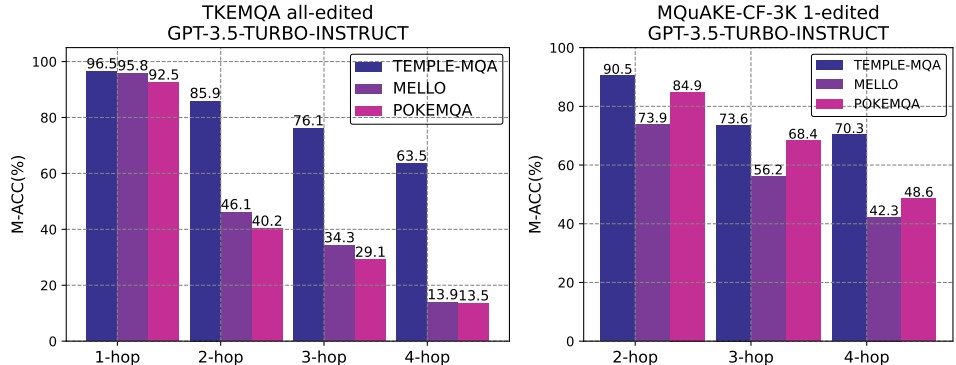

Figure 5: A bar graph comparing the M-Acc of three methods under two settings: All-edited for TEMPLE-MQA and 1-edited for MQUAKE-CF-3K.

the inference path, while Mello and PokeMQA need to call LLMs multiple times to generate sub-questions. Thus, we claim TEMPLE-MQA offers a faster and more cost-effective solution.

**Accuracy of Structured Retrieval.** To achieve reliable KE, apart from the correct decomposition of sub-problems, another important factor is whether related edits can be found correctly. Here, we conduct experiments to compare the accuracy of the retrieval process for different methods. According to Table 14, TEMPLE-MQA achieve higher retrieval accuracy compared to Mello and PokeMQA by 17.9% and 7.7% in the all-edited setting, respectively. Thus, our structured retrieval can efficiently find structured edits.

**TEMPLE-MQA under Different Hop Numbers.** Here, we aim to study the performance of different models with different hop numbers. As shown in Figure 5, TEMPLE-MQA can still maintain a high M-Acc as the number of hops increases, while Mello and PokeMQA experience an obvious decline in performance with an increase in hop number. This is because Mello and PokeMQA couple the tasks of querying the LLM to answer sub-questions and generate sub-questions together. As the hop number increases, the LLM is burdened with complex tasks, causing a significant decrease in M-Acc. In contrast, TEMPLE-MQA decouples the generation of inference paths, enabling it to maintain high performance as the hop number increases.

**TEMPLE-MQA under Implicit Time.** For questions with hidden time, we consider the following examples: 1) "Who was the U.S. president before Biden?", 2) "Who was the previous president of the U.S.?".

For example 1, TEMPLE-MQA needs to determine the specific time when Biden is president. We can revise the ICL prompt to achieve it, as show in below:

Question: Who was the U.S. president before Biden?
Time: Biden is president after 2021, Thus time span is <2017,2020>.
Inference Path: <U.S.,president is,?>.

However, we observe some datasets contain questions that are more ambiguous, as shown in examples 2. Answer to these questions is decided on the training periods of LLMs. For this, we need to give current time to LLM, as shown in below:

Question: Who was the previous president of the U.S.?
Time: Today is 2021, and Biden is current president, Thus time span is <2017,2020>.
Inference Path: <U.S.,president is,?>.

The results on the AToke-SE and AToke-ME with ambiguous hidden time (like example 2) are shown in Table 12. We can see that TEMPLE-MQA still performs better, especially on the HRS metric.

| | DATA 998 IN MQUAKE-CF-3K |
|---|---|
| **Questions** | What continent does Gareth David-Lloyd hold citizenship in?
In which continent was the actor Gareth David-Lloyd a citizen of?
Which continent does the citizenship of Gareth David-Lloyd belong to? |
| **Edits** | (Gareth David-Lloyd, country of citizenship, Nigeria)
(Nigeria, continent, North America) |
| | **DATA 706 IN MQUAKE-CF-3K** |
| **Questions** | What continent does the country Peter Green belong to citizenship to belong to?
Which continent is the country of citizenship of Peter Green?
On which continent is the country where Peter Green holds citizenship located? |
| **Edits** | (Peter Green, country of citizenship, Nigeria) |
| | **ORIGINAL INFERENCE PROCEDURE IN DATA 706** |
| **Inference procedure** | (Peter Green, country of citizenship, Nigeria)
(Nigeria, continent, Africa) |
| **Original Answer** | Africa |
| | **DISTURBED INFERENCE PROCEDURE IN DATA 706** |
| **Inference procedure** | (Peter Green, country of citizenship, Nigeria)
(Nigeria, continent, North America) |
| **Wrong Answer** | North America |

Table 15: **Data Conflict in MQUAKE-CF-3K.** Under the setting of mass editing, a step (highlighted in blue) in the original inference procedure in Date 706 is changed due to retrieving an edit in Data 998 (highlighted in red), resulting in a wrong answer.

## D Temporal Knowledge Editing for Multi-hop Question Answering Benchmark: TKEMQA

| | QUESTIONS |
|---|---|
| **Current question** | In which continent is the country where the head of government of Taiwan has citizenship located after 2023? |
| **Historical questions** | Who is the head of government in Taiwan from 2019 to 2023?
Who is the head of government in Taiwan from 2017 to 2019? |
| | **KNOWLEDGE STORE IN MEMORY** |
| **Corrective knowledge** | (Chen Chien-jen, country of citizenship, Taiwan → Algeria) |
| **Updated knowledge** | (Taiwan, head of government, Chen Chien-jen, 2023, N/A) |
| **Historical knowledge** | (Taiwan, head of government, Hope Su, 2019, 2023)
(Taiwan, head of government, Lai Ching-te, 2017, 2019) |
| | **PROCEDURE TO ANSWER CURRENT QUESTION** |
| **Inference procedure** | (Taiwan, head of government, Chen Chien-jen)
(Chen Chien-jen, country of citizenship, Algeria)
(Algeria, continent, Africa) |
| | **ANSWERS** |
| **Current answer** | Africa |
| **Historical answers** | Hope Su
Lai Ching-te |

Table 16: **An instance in TKEMQA-HK.** Updated knowledge refers to the updating of outdated knowledge and does not deny the correctness of historical knowledge. Corrective knowledge refers to the subversion of a model's past knowledge used to simulate the model's modification of erroneous knowledge. The updated knowledge and corrective knowledge required in the inference procedure are highlighted in red and blue, respectively.

### D.1 Introduction to TKEMQA

Our benchmark TKEMQA is created based on Wikidata [2], a knowledge base consisting of fact triples associated with millions of entities. TKEMQA can be used to evaluate whether the knowledge editing method can update new knowledge without forgetting past knowledge. It contains simple single questions and multi-hop questions, and its rich settings enable it to comprehensively evaluate knowledge editing methods.

As shown in Table 9, TKEMQA contains 2,000 data where each one includes a k-hop ($k \in \{1, 2, 3, 4\}$) questions. As GPT-3.5 does not have the knowledge after 2022, we use the knowledge after 2022 as an edit to *update knowledge*. Thus, we refer to each multi-hop question as a current question because it has a time prompt after 2022. Besides the updated knowledge, we also introduce the *corrective knowledge*, which is used to modify the error historical knowledge. This knowledge does not need temporal scope and is counterfactual.

To evaluate whether the editing method may cause model forget the historical knowledge (i.e. knowledge of model before the editing), each data of TKEMQA include two historical question with the time prompt before 2022. Based on TKEMQA, we add the knowledge corresponding to the historical question into the edit memory to evaluate whether the retrieval can effectively distinguish different knowledge in the same time chain, in which we refer to as TKEMQA-HK.

We use M-Acc and HES as two metrics in TKEMQA. M-Acc is the accuracy of the language models on current multi-hop questions, evaluating the model's ability to update knowledge and perform multi-hop reasoning successfully. HES is the accuracy of answering historical questions and assessing the model's ability to recall historical knowledge.

### D.2 TKEMQA Data Curation

**Collect relation template.** We first manually collect multiple relations containing time information from WikiData, e.g. "President is" (Appendix D.3). Then, we use SPARQL to sample facts based on the relation and subject entity. For example, given a query (U.S., President is, ?), WikiData will return every president of the U.S. and their terms of office. In addition, we select common relations and entities to control quality.

**Generate relation path template.** In order to generate natural and coherent multi-hop questions, we look for high-quality relation path Luo et al. (2023) with different lengths $k \in \{1, 2, 3, 4\}$. (live in, the father is) is a bad path because the object type of 'life in' (location) does not match the subject type of 'father is' (person); a location cannot have a father. Thus, we arrange and combine the collected relations into a relation path template and filter out unreasonable relation combinations. For each relation path, we can get several fact chains based on different start point entities.

**Generate multi-hop questions.** Then, we use GPT-4 to convert the relation path template into questions in natural language form, e.g., (father is, company is) -> "What is the company of father of {}?". Replace "{}" into the start point entity to get the final questions. We also conduct manual revisions to ensure its quality. Different from existing work (Zhong et al., 2023), this method produces data with higher quality questions due to the powerful capabilities of GPT-4. This method also helps reduce the additional overhead of using GPT-4 because one path often correspond to multiple data items.

**Additional details for edits.** Among the collected fact chains, there is only one piece of knowledge that is after 2023, which is used to represent the knowledge update of the real world.

### D.3 Question/Cloze Statement Templates used in TKEMQA

Following previous work, we use question templates to filter the facts that cannot be recalled and cloze-style statement templates to convert an edited fact into a natural language form

---

[2] https://www.wikidata.org/wiki/Wikidata:Main_Page

statement. Table 17 shows question templates and cloze-style statement templates employed in TKEMQA.

| Relation | Question template | Cloze-style statement template |
|---|---|---|
| P35 | Who is the head of state in [S]? | The head of state in [S] is |
| P6 | Who is the head of government in [S]? | The head of government in [S] is |
| P488 | Who is the chairperson of [S]? | The chairperson of [S] is |
| P169 | Who is the chief executive officer of [S]? | The chief executive officer of [S] is |
| P54 | Which sports team is [S] affiliated with? | [S] is affiliated with the sports team of |
| P286 | Who is the head coach of [S]? | The head coach of [S] is |
| P551 | Where does [S] live? | [S] live in |
| P102 | Which political party is [S] affiliated with? | [S] is affiliated with the political party of |
| P26 | Who is [S]'s spouse? | [S]'s spouse is |
| P38 | What is the currency of [S]? | The currency of [S] is |
| P108 | Which organization is [S] an employee of? | [S] is an employee of |
| P69 | Which university is [S] educated at? | [S] is educated at |
| P937 | Where is [S]'s workplace? | The work location of [S] is |
| P36 | What is the capital of [S]? | The capital of [S] is |
| P159 | Where is the headquarters of [S] located? | The headquarters of [S] is located in |
| P27 | What is the country of citizenship of [S]? | [S] is a citizen of |
| P140 | Which religion is [S]affiliated with? | [S] is affiliated with the religion of |
| P30 | Which continent is [S] located in? | [S] is located in the continent of |
| P37 | What is the official language of [S]? | The official language of [S] is |
| P17 | Which country is [S] located in? | [S] is located in the country of |

Table 17: Question templates and cloze-style statement templates employed in TKEMQA. "[S]" denotes a placeholder for the subject entity within the fact. The question templates filter the facts that cannot be recalled. The cloze-style statement templates convert an edited fact into a natural language form statement.

## E  SPARQL Protocol and RDF Query Language

SPARQL is able to retrieve and manipulate data stored in Resource Description Framework (RDF) format, which can store graph information. Wikidata Query Service (WDQS) is an online tool for querying the Wikidata database. It allows users to retrieve and analyze structured data in Wikidata using the SPARQL query language. We use SPARQL to access WDQS to obtain the alias of the subject entity and create our dataset TKEMQA (see the table 18).

| | Using SPARQL to extract aliases |
|---|---|
| **SPARQL** | SELECT ?alias WHERE {
    wd:<QID> skos:altLabel ?alias.
    FILTER(LANG(?alias) = "en").
} |
| **Description** | This SPARQL query retrieves English alias for a specific entity. "<QID>" denotes a placeholder for the entity id in Wikidata. |
| | **Using SPARQL to extract facts** |
| **SPARQL** | SELECT ?x ?Label WHERE {
    wd:<QID> wdt:<PID> ?x.
    ?x rdfs:label ?Label.
    FILTER(LANG(?Label) = "en")
} |
| **Description** | This SPARQL is used to retrieves corresponding object entity according to the subject entity id (represented by "<QID>") and relation id (represented by "<PID>"). |
| | **Using SPARQL to extract facts with temporal scope** |
| **SPARQL** | SELECT ?x ?Label ?start_time ?end_time WHERE {
    wd:<QID> p:<PID> ?statement.
    ?statement ps:<PID> ?x.
    ?x rdfs:label ?Label.
    OPTIONAL{ ?statement pq:P580 ?start_time.}
    OPTIONAL{ ?statement pq:P582 ?end_time.}
    FILTER(LANG(?Label) = "en").
}
ORDER BY DESC (?start_time) |
| **Description** | This SPARQL query is used to extract object entities and their corresponding time scopes based on subject entity id (denoted by "<QID>") and relation id (denoted by "<PID>"). |

Table 18: **The SPARQL we used to create TKEMQA.**

| Unstructured Form | Structured Form |
|---|---|
| From 1993 to 1999, Donald Trump's spouse is Marla Maples. | (Donald Trump, spouse, person, 1993, 1999, Marla Maples) |
| In 2020, the head coach of AC Horsens is Jonas Dal. | (AC Horsens, head coach, person, 2020, 2020, Jonas Dal) |
| After 2023, the head of government in Slovakia is Robert Fico. | (Slovakia, head of government, person, 2023, N/A, Robert Fico) |
| Before 1936, the head of state in United Kingdom is George V. | (United Kingdom, head of state, person, N/A, 1936, George V) |
| The capital of United Kingdom is London. | (United Kingdom, capital, city, N/A, N/A, Londo) |

Table 19: Examples of transforming knowledge from unstructured form into structured form. The N/A indicate unknown for start time or end time..

