# OpenReview forum: "Multi-hop Question Answering under Temporal Knowledge Editing"
_colmweb.org/COLM/2024/Conference — COLM_

### Official Review · Reviewer_Er5T · 2024-05-11

**Rating:** 6
**Confidence:** 4
**Ethics Flag:** 1

**Summary:**

The paper proposes a novel framework called TEMPLE-MQA (TEMPoral knowLEdge augmented Multi-hop Question Answering) for multi-hop question answering under temporal knowledge editing. TEMPLE-MQA constructs a time-aware graph (TAG) to store knowledge edits in a structured manner, preserving temporal information. It utilizes pre-trained language models (LLMs) to devise an inference path and perform joint reasoning using the TAG and LLMs. The key contributions include the TAG construction method, inference path generation, structural retrieval for knowledge retrieval, and a new dataset called TKEMQA for evaluating temporal MQA. Experimental results on benchmark datasets demonstrate that TEMPLE-MQA outperforms existing baselines for MQA under massive knowledge edits.

**Questions To Authors:**

1. In Section 4.1 on constructing the Time-Aware Graph (TAG), you mention using data augmentation to capture aliases for entity names to aid entity disambiguation. Could you provide some more details on how this data augmentation is performed and how effective it is compared to not using data augmentation?

2. The results in Table 2 show that on TKEMQA, TEMPLE-MQA underperforms MeLLo on the HES metric in the All-edited GPT setting. What are some potential reasons for this and are there any planned improvements to address it?

**Reasons To Accept:**

1. The paper addresses an important problem of handling temporal contexts in multi-hop question answering with knowledge editing, which has not been well-explored in previous work.
2. The introduction of the TKEMQA dataset, the first benchmark tailored for MQA with temporal scopes, is a valuable contribution to the research community.
3. Extensive experiments on multiple datasets demonstrate the superior performance of TEMPLE-MQA over existing baselines, validating the effectiveness of the proposed approach.
4. The paper is well-written, with clear explanations of the methodology and experimental setup.

**Reasons To Reject:**

1. While the paper introduces the TKEMQA dataset, more details on the dataset construction process and the quality of the data could be provided.
2. The paper could benefit from more extensive ablation studies and analyses to better understand the contributions of different components in TEMPLE-MQA.
3. The computational complexity and scalability of TEMPLE-MQA, especially for large-scale knowledge graphs, are not thoroughly discussed.
4. The paper lacks discussions on the limitations of the proposed approach and potential future directions.

---

> ### Author Rebuttal · Authors · 2024-05-31
>
> Thanks for your valuable feedback. Our response for each question is as follows:
>
> **W1. Data Quality**
>
> Please refer to our response to reviewer-EvWU.
>
> **W2. Ablation Studies:**
>
> We performed two ablation experiments on MQUAKE:
>
> | Method                 | 1-edited | 100-edited | all-edited |
> | ---------------------- | -------- | ---------- | ---------- |
> | TEMPLE-MQA             | 78.11    | 67.21      | 53.68      |
> | -w/o data augmentation | 74.54    | 63.81      | 50.67      |
> | -w/o inference path    | 62.67    | 52.82      | 44.33      |
>
> After removing data augmentation, the M-ACC drops by up to 3.57%. We test TEMPLE-MQA with sub-question decomposition excluding the inference path. the M-ACC drops by up to 15.44%. It has a huge impact on performance.
>
> **W3. Scalability**
>
> We perform a quick empirical analysis on large KG (X represents the multiple of all the original edits):
>
> | Size          | 1x    | 2x    | 4x    | 10x   |
> | ------------- | ----- | ----- | ----- | ----- |
> | Acc (%)       | 58.67 | 56.49 | 54.31 | 50.67 |
> | Runtime (min) | 326.1 | 328.9 | 336.8 | 346.7 |
>
> These results show that Temple-MQA is scale-able. We will add these details to the final draft to clarify the computational complexity and scalability of our work.
>
> **W4. Limitations**
>
> Our work poses following limitations:
>     * Temple-MQA requires pre-processing of TAG, which brings additional overhead.
>     * Temple-MQA is very effective on the existing and our proposed benchmarks, but it requires writing prompts manually to adapt to different tasks.
>     * Although retrieval of Temple-MQA doesn't need fine-tuning an encoder, it has a complex workflow.
>
> **Q1. Disambiguation:**
>
> Entity disambiguation is a core part of our model, as it helps improve retrieval performance by filtering entities that are not relevant for a given context. We report the results of our model with and without data augmentation in the table given above. The performance declines by up to 3.57%.
>
> **Q2. Strange results**
>
> The possible reason for the performance shortfall is that the TEMPLE-MQA uses a structured query prompt:
> ```
> Inference step: <U.K., head of government, person>
> Time: <2019,2022>
> Answer: Boris Johnson
> ```
> which leads to lower accuracy than sub-questions with natural language form adopted in MeLLo:
>
> ```
> Question:Who was the head of government of U.K. from 2019 to 2022?
> Answer: Boris Johnson
> ```
> One possible way to address this problem is using natural language form query prompts.

---

> > ### Comment · Reviewer_Er5T · 2024-06-06
> >
> > Thanks for the response and explanation! It will be helpful to add the explanation into the revision.

---

### Official Review · Reviewer_huoN · 2024-05-11

**Rating:** 8
**Confidence:** 4
**Ethics Flag:** 1

**Summary:**

The paper proposes a novel method to multi-hop question answering especially focusing on how to correctly answer temporal questions. The authors use a temporal aware graph (TAG) representing temporal information. The graph is used to retrieve the correct answer to temporal questions. Comparison with previous approaches shows superiority of the new method for two benchmark datasets. In addition, a new data set is proposed.

**Questions To Authors:**

While the overall contribution is definitely novel and the results are impressive, I found it difficult to understand the different steps of the algorithm that was developed. Sometimes the authors referred to the appendix and some of the explanations seemed confusing and  incomplete:

- 4.1. Comparison to existing approaches:”..store knowledge edits in an unstructured format.. “ please give an example. Does unstructured format simply mean text?
- “Historical  and current facts are typically stored in unstructured format  as f_ols and f_current” this sentence does not add much to the explanation

- “See appendix A for details.” PLese add more information in the text because this is an important part of your paper

- Data augmentation (SPARQL) and graph construction refer to the appendix again and the text lacks clarity. It may be helpful to include a running example so that the reader can see the effects of the different steps.

- “alternate between multiple different and solve phases. “ noun missing after different?

- Evaluation metrics: the metrics are specifically developed for the temporal Multi-hop QA and it would be good to describe them in more detail in the paper because otherwise the reader will have a hard time identify the significance of the difference in performance (i.e., Historical Explicit time Question Score (HES) and Current Explicit time Question Score (CES) )
- Joint reasoning: “there is no need to use self- checking Zhong et al. (2023) to check if the retrieved fact contradicts the generated answer. “ why?

**Reasons To Accept:**

-	Novel extension of LLM-based Mulit-hop question answering using temporal information
-	The representation of a temporal graph indicates an important step towards reasoning capabilities of temporal information
-	The new method shows strong performance compared to previous approaches

**Reasons To Reject:**

The paper is sometimes difficult  to follow, several important aspects are hidden in the appendix or refer to other papers:
- Construction of the temporal graph generation
- the metrics used.

---

> ### Author Rebuttal · Authors · 2024-05-31
>
> Thanks for your valuable feedback. Our response for your questions is as follows:
>
> **Response to Reasons**
>
> W1. See our response to Question 2,3.
>
> W2. See our response to Question 6.
>
> **Response to Questions**
>
> **Q1. Unstructured:**
>
> Unstructured format means sentences in natural language form. For example, "The president of the U.S. is Joe Biden." is an unstructured text format, while (U.S., president is, Joe Biden) is structured. In the future, we plan to extend it to support knowledge editing by complex unstructured paragraphs. E.g., "The Eiffel Tower is located in Paris. It was designed by french engineer Gustave Eiffel...", we can extract knowledge (Eiffel Tower, located in, Paris),(Eiffel Tower, design by, Gustave Eiffel),(Gustave Eiffel, nationality, French).
>
> **Q2. f\_old and f\_cur:**
>
> We add two examples to better explain it: f\_old means historical knowledge, e.g. "The president of the U.S. was Donald Trump from 2016 to 2021." f\_cur means current knowledge, e.g., "After 2021, the President of the U.S. is Joe Biden."
>
> **Q3. Appendix-A:**
>
> We will augment the details in Appendix-A. Also, we explain the contents of Appendix A as follows:
>         - We add the key features of inference-path planning prompt to support high accuracy, such as amount and diversity of demos.
>         - We give some running examples to make the workflow easier for the readers.
>
> **Q4. Data Augmentation**
>
> We explain it through a simple example: For entity 'United States of America'(its id is Q30), we first construct the following SPARQL statement:
> ```
> SELECT ?alias WHERE {
> wd:Q30 skos:altLabel ?alias.
> FILTER(LANG(?alias) = "en").
> }
> ```
> Then we query the website "query.wikidata.org/" to get all the aliases (USA, United States, the US ...).
>
> **Q5. Typo:**
>
> We rephrase it as "alternate between plan and solve phases".
>
> **Q6. Metrics:**
>
> Owing to limited space, we provided details about the evaluation metrics in Appendix B.3.
>
> **Q7. Self-checking:**
>
> Self-checking implies using LLMs to check if the retrieved fact contradicts the generated answer. The reason why we do not use self-checking is that we have observed that many LLMs have been trained with RLHF, which makes them have a high tendency to reject external knowledge that contradicts their own knowledge. However, we admit that self-checking is a very good design that can effectively resist harmful edits.

---

> > ### Comment · Reviewer_huoN · 2024-06-04
> >
> > thank you for the explanations/comments. I think the suggested changes will improve clarity of the paper.

---

### Official Review · Reviewer_EvWU · 2024-05-11

**Rating:** 6
**Confidence:** 3
**Ethics Flag:** 1

**Summary:**

The proposed method introduces a novel approach for handling multi-hop question answering that involves substantial volumes of temporal knowledge edits. The proposed approach consists of 1) extending a fact triple knowledge graph with timestamps (referred to as a time-aware graph or TAG by the authors). Such structured information can then be converted into unstructured text by an LLM for Data Augmentation and Contextual Filtering to enhance retrieval performance. Given these improvements, they also propose using LLM and TAG for chain-of-thought type of step-by-step joint reasoning.

**Reasons To Accept:**

- The proposed approach addresses an interesting and challenging problem on multiple-hop QA with temporal time stamps.
- Expanding the knowledge graph with time stamps is an interesting direction to explore.
- The authors develop a temporal multi-hop QA dataset that can benefit the community.
- The proposed approach is novel, although seems to stitch multiple different existing papers.

**Reasons To Reject:**

- My main concern is with the evaluation. Are there quality controls on the created TKEMQA dataset?
 In section 3, the temporal KE from start time to end time is added to the original fact triple. However, it was not clear to me how reliable the augmented facts are. Are these being evaluated? It seems that they are only evaluated on downstream multi-hop QAs with accuracy.
- There are some recent papers on temporal multi-hop QA datasets that are worth comparison to the differences. e.g. PAT-Questions: A Self-Updating Benchmark for Present-Anchored Temporal Question-Answering https://arxiv.org/abs/2402.11034

---

> ### Author Rebuttal · Authors · 2024-05-31
>
> Thanks for your valuable feedback. The response to each question is as follows:
>
> **W1.  Datasets**
>
> We further elaborate on our data construction process and associated quality control, as follows:
>
>  * **Sampling Fact Triplets.** We first manually collect multiple relations containing time information from WikiData, e.g. "President is". Then, we use SPARQL to sample facts based on the relation and subject entity. For example, given a query (U.S., President is, ?), WikiData will return every president of the U.S. and their terms of office. Thus, We select common relations and entities to control quality.
>
> * **Generate Relation Path Template.** In order to generate natural and coherent multi-hop questions, we look for high-quality relation combinations. E.g., (live in, the father is) is a bad combination because the object type of 'life in' (location) does not match the subject type of 'father is' (person); a location cannot have a father. Thus, we arrange and combine the collected relations into a relation path template and filter out unreasonable relation combinations.
>
> * **Generate Multi-hop Question Template:** Then, we use GPT-4 to generate the multi-hop question template for each relation path template, e.g., (father is, company is) -> "What is the company of father of {}?". Replace "{}" into the start point entity to get the final questions. We also conduct manual revisions to ensure its quality.
>
> * **Augmented data is reliable:** All the augmented facts are reliable and are sampled from Wikidata after 2022.
>
> We will add these details in the final draft to further clarify the data curation pipeline of our work.
>
> **W2. Compare with other papers**
>
> There are some recent papers on temporal multi-hop QA datasets that are worth comparison to the differences.
>
> * Thank you for pointing out this related work. We noticed that the paper you have mentioned only covers temporal MQA and may not be well-suited for temporal MQA under KE. Moreover, the paper was released one and a half months before the CoLM deadline. However, we will try to add its comparisons for inference path planning against Temple-MQA in the final draft.

---

> > ### Comment · Reviewer_EvWU · 2024-06-03
> >
> > Thanks for your response. I will stick with my original rating.

---

### Official Review · Reviewer_JhSs · 2024-05-12

**Rating:** 7
**Confidence:** 4
**Ethics Flag:** 1

**Summary:**

This paper proposed a temporal knowledge augmented multi-hop question answering method called Temple-MQA. This method built a time-aware graph to store edits as a time-conscious knowledge structure. The benefit of this graph is to store the temporal information in a structured format. During the retrieval, Temple-MQA utilized a LLM to devise a path and do a step-by-step reasoning. It is interesting to use LLM to strategize an inference path. In general, the whole idea is clear with a reasonable performance.

**Questions To Authors:**

Look forward to more analysis on various temporal question types.

**Reasons To Accept:**

Strengths:

1.	This paper presented a temple-MQA method that achieves a high accuracy for multi-hop temporal QA task. The whole architecture includes the following key steps: 1) construct a time-aware graph (TAG) to store time information; 2) build an inference path for each multi-hop question using LLMs; 3) design a joint reasoning way to combine LLMs with TAG.

2.	The performance of proposed temple-MQA achieves the state of art performance on multiple metrics. It proves the effectiveness of proposed method.

**Reasons To Reject:**

Weaknesses:

1.	The temporal questions include two types: questions with explicit time (like 01/01/2024) and questions with hidden time (like before, after). It will be wondering if more deep analysis cross different types are provided. The questions with hidden time will require more support about multi-step reasoning. Maybe this method can perform well on various types.

2.	Using the LLMs to generate the inference path is a great idea. But this section missed some details. I would like to see more comparisons and analysis about this direction. The accuracy of inference path will impact the performance a lot.

---

> ### Author Rebuttal · Authors · 2024-05-31
>
> Thanks for your valuable feedback. We address your concerns as follows.
>
>  **W1. Implicit Time**
>
> For questions with hidden time, we consider the following examples:
>
> * "Who was the U.S. president before Biden?",
> * "Who was the previous president of the U.S.?"
>
> For example 1, TEMPLE-MQA needs to determine the specific time when Biden is president. We can revise the ICL prompt to achieve it. For example,
>
> ```
> Question: Who was the U.S. president before Biden?
> Time: Biden is president after 2021, Thus time span is <2017,2020>.
> Inference Path: <U.S.,president is,?>
> ```
>
> However, we observe some datasets contain questions that are more ambiguous, as shown in examples 2. Answer to these questions is decided on the training periods of LLMs. For this, we need to give current time to LLM.
>
> ```
> Question: Who was the previous president of the U.S.?
> Time: Today is 2021, and Biden is current president, Thus time span is <2017,2020>.
> Inference Path: <U.S.,president is,?>
> ```
>
> The results on the AToke-SE and AToke-ME with ambiguous hidden time (like example 2) are shown below:
>
> | Method     | CRS(SE) | HRS(SE) | CRS(ME) | HRS(ME) |
> | ---------- | ------- | ------- | ------- | ------- |
> | MEND+      | 25.41   | 30.17   | 21.84   | 30.83   |
> | ROME+      | 78.88   | 16.29   | 82.40   | 18.18   |
> | MEMIT+     | 74.07   | 24.32   | 73.58   | 26.04   |
> | TEMPLE-MQA | 85.81   | 78.23   | 87.74   | 80.92   |
>
> We can see that TEMPLE-MQA still performs better, especially on the HRS metric.
>
> **W2. Inference Path**
>
> For this, we add two experiments:
>     * Ablation Study: remove inference path planning.
>     * Explore the influence of the number of shots on the ICL prompt.
>
> Here are our results
>
> | Method              | 1-edited | 100-edited | all-edited |
> | ------------------- | -------- | ---------- | ---------- |
> | TEMPLE-MQA(8-shot)  | 78.11    | 67.21      | 53.68      |
> | -w/o inference path | 62.67    | 52.82      | 44.33      |
> | 1-shot              | 65.71    | 55.12      | 46.87      |
> | 2-shot              | 69.48    | 58.72      | 47.48      |
> | 4-shot              | 73.93    | 64.63      | 49.91      |
>
> If we exclude the inference path, the M-ACC drops about 15.44%, 14.39%, 9.53% respectively. The M-ACC are drops about 10.43%, 7.77% and 3.51% in 1,2 and 4 shot setting. Thus more examples can help LLMs follow instructions better, also the quality and diversity of the examples is important.
>
> We will add the above findings in the final version.

---

> > ### Comment · Reviewer_JhSs · 2024-06-05
> > **Re: Rebuttal by Authors**
> >
> > Thanks a lot for providing more results and clarifying the questions. It is pretty interesting about the implicit time analysis. I would like to recommend to add it into the final version. In general, the whole paper looks great. I'd like to raise my rating.

---

### Decision · Program_Chairs · 2024-07-10

**Decision:**

Accept

**Comment:**

This paper proposed to use time-aware graph to store time-conscious knowledge, which is used for answering temporal multi-hop questions.

Pros:
1. The problem studied is important.
2. The method achieves high accuracy.
3. Some reviewers commended the novelty of the method.

Cons:
1. Reviewers asked for elaboration of some details.
2. Reviewers also asked for some additional ablation and comparison with existing work to strengthen the evaluation.

The authors addressed some of reviewers’ concerns in the rebuttal. Overall, all four reviewers are positive about this paper. There are not any critical concerns. The authors are encouraged to include the additional results and elaboration in their final version of the paper if the paper is accepted.